# OpenReview forum: "CodePDE: An Inference Framework for LLM-driven PDE Solver Generation"
_TMLR — Accepted by TMLR_

### Review · Reviewer_vX6Y · 2025-12-03

**Summary Of Contributions:**

## Summary
This paper proposes CodePDE, an inference framework for generating numerical PDE solvers using LLMs. The framework consists of five stages—task specification, code generation, automated debugging, evaluation, and solver refinement—and is applied to 16 LLMs across five representative PDE families.

The authors further analyze model behaviors including solver correctness, numerical scheme diversity, convergence properties, and the effects of test-time scaling. The work offers several interesting empirical insights, such as the trade-off between solver reliability and solver sophistication, and differing strengths across reasoning-oriented vs refinement-oriented LLMs.

## Strengths
1. Framing PDE solving as LLM-driven code generation is a fresh and potentially impactful perspective. This paradigm opens useful avenues for automation, interpretability, and hybrid symbolic–numerical reasoning in scientific computing.
2. The experiments cover 16 prominent LLMs, 5 PDE families with varying difficulty and characteristics, Multiple metrics, and Test-time scaling curves. This is a very comprehensive evaluation of LLM-based PDE solver.
3. The CodePDE pipeline is clear, modular, and pragmatic. The inclusion of iterative debugging and refinement significantly increases solver reliability and performance, demonstrating realistic agentic capabilities.

## Weaknesses
1. While useful, the framework largely combines generic inference techniques—prompting, best-of-n sampling, iterative debugging, and LLM refinement—without introducing PDE-specific algorithms or new inference methods.
2. Missing ablations on PDE-specific prompting or model knowledge. It would be highly informative to explore: (1) Providing PDE textbooks or numerical methods summaries to the model (2) Enforcing numerical constraints (CFL, positivity, mass conservation) within prompts (3) The impact of adding symbolic tools
3. Some key components lack precise algorithmic specification. For example, The refinement stage does not clearly describe how the nRMSE feedback is structured for the model.

**Audience:**

Yes

**Audience Explanation:**

N/A

**Claims And Evidence:**

Yes

**Claims Explanation:**

N/A

**Requested Changes:**

See weaknesses.

---

### Review · Reviewer_nLdX · 2025-12-03

**Summary Of Contributions:**

This paper introduces CodePDE, a new inference-time framework that uses large language models (LLMs) to automatically generate, debug, and refine numerical solvers for partial differential equations (PDEs).

Instead of training specialized neural PDE solvers, CodePDE treats PDE solving as a code-generation problem: given a natural-language description of a PDE and its boundary/initial conditions, an LLM writes executable solver code (e.g., finite-difference, spectral, or IMEX methods).

The framework adds three key inference-time components that make this process practical and reliable:

Automated Debugging — runs generated code, captures runtime errors, and prompts the LLM to fix bugs iteratively.

Self-Refinement — uses numerical feedback (nRMSE) to improve solver accuracy through feedback-driven rewriting.

Test-Time Scaling — employs best-of-n sampling to enhance solver quality by leveraging additional inference compute.

Across five canonical PDE families (Advection, Burgers, Reaction–Diffusion, Compressible Navier–Stokes, and Darcy Flow), and 16 different LLMs, CodePDE produces high-quality, convergent solvers — often matching or exceeding classical hand-crafted methods and outperforming existing neural solvers that require expensive offline training.

The work also presents a systematic analysis of LLM behaviors in scientific programming, uncovering:

a trade-off between solver reliability (safe, low-order schemes) and sophistication (high-order or diverse methods);

distinct model capabilities for generation versus refinement; and

clear evidence that debugging + refinement + scaling are the essential ingredients for any scientific “agentic” workflow.

Finally, because the generated solvers are human-readable, CodePDE enables transparent error diagnosis and interpretability — offering a practical bridge between symbolic scientific reasoning and automated code synthesis.

**Audience:**

Yes

**Audience Explanation:**

Relevance to TMLR Audience
1. Strong alignment with machine learning research interests

TMLR (Transactions on Machine Learning Research) targets readers who care about:

Advances in learning paradigms (not only model training but also inference, reasoning, and agentic workflows),

Intersections of ML with scientific domains, and

Empirical rigor and reproducibility.

CodePDE fits squarely within these aims because it:

Extends ML into scientific reasoning and simulation, a rapidly expanding area (ML for science).

Proposes a novel inference framework rather than yet another model architecture — an idea-driven contribution consistent with TMLR’s scope.

Provides extensive, reproducible experiments across 16 LLMs and 5 PDE families.

So it would interest readers who follow both scientific ML and LLM reasoning trends.

2. Bridges two active research communities

Scientific Machine Learning / Physics-Informed ML: Researchers developing PINNs, FNOs, or operator-learning methods would be intrigued that inference-only LLMs can sometimes outperform trained neural PDE solvers.

LLM Reasoning & Agentic Systems: Those studying chain-of-thought, self-refinement, and test-time scaling will see this as a concrete, quantitative domain where those mechanisms demonstrably matter.

This bridge between symbolic scientific computing and LLM reasoning is rare — and therefore valuable to a broad TMLR audience.

3. Provides empirically grounded insights, not hype

The study contributes:

Rigorous benchmarks (PDEBench, FNO datasets),

Quantitative analyses (nRMSE, convergence orders),

Ablation results demonstrating which inference-time components truly matter.

TMLR reviewers and readers appreciate clear evidence over novelty for its own sake, and this work delivers that.

**Broader Impact Concerns:**

The ethical risks associated with this work are relatively low, as it focuses on automated code generation for scientific computing rather than on human data or decision-making. However, a few broader-impact aspects could be better addressed. First, since the framework executes LLM-generated code, there are potential safety and security concerns (e.g., malicious or unstable code execution, unbounded resource use) that merit explicit discussion of sandboxing, auditing, or containment strategies. Second, the environmental cost of extensive inference-time sampling (best-of-n and multi-round refinement) could be acknowledged, as repeated LLM calls substantially increase compute usage. Finally, the authors might comment briefly on the reliability and verification of automatically generated scientific code if such methods are adopted in critical applications (e.g., engineering or climate modeling).

These are not severe ethical shortcomings, but adding a short Broader Impact paragraph addressing code-execution safety, computational sustainability, and responsible deployment of automated scientific solvers would strengthen the submission.

**Claims And Evidence:**

Yes

**Claims Explanation:**

🧩 1. Claim:

LLMs can generate high-quality PDE solvers that rival or outperform traditional and neural solvers.

Evidence:

Quantitative comparisons in Table 1 show lower normalized RMSE (nRMSE) for LLM-generated solvers (with refinement) on 4 of 5 PDE families.

Example: Burgers Equation — Gemini 2.0 Flash achieves
1.06
×
10
−
4
1.06×10
−4
 vs
3.55
×
10
−
4
3.55×10
−4
 for the reference solver.

Neural baselines (PINN, FNO, UPS foundation model) perform worse despite training.

Assessment: ✅ Convincing

The use of multiple PDEs and baselines supports the claim well.

The accuracy metric (nRMSE) and convergence tests are appropriate.

Minor caveat: reference solver tuning is not deeply discussed, but the performance margin is large enough to be credible.

🧩 2. Claim:

Structured inference — combining debugging, refinement, and test-time scaling — is essential for reliable solver generation.

Evidence:

Section 5.2 → 5.4 + 5.6 (Ablation Study) directly quantify this.

Debugging boosts executable-code success from 41 % → 84 %.

Refinement converts underperforming solvers into surpassing references (e.g., CNS 2.41 × 10⁻² → < 1.89 × 10⁻²).

Best-of-n scaling improves accuracy up to n ≈ 16.

Removing these components (naïve prompting) yields unusable nRMSE ≈ 1.5 × 10⁻¹.

Assessment: ✅✅ Very convincing and quantitatively backed.

Clear causal link between framework components and outcome.

Multiple metrics (nRMSE, convergence, success rate) triangulate the effect.

🧩 3. Claim:

There exists a trade-off between solver reliability and sophistication (simple vs. high-order schemes).

Evidence:

Section 5.5 Convergence Analysis:

GPT-4o mini → low failure (8.6 %) but first-order methods.

Gemini 2.5 Pro and o3 → higher-order methods but more failures.

The authors visualize convergence-order distributions across LLMs.

Assessment: ✅ Supported and well interpreted.

Quantitative data plus qualitative explanation (safety vs. exploration) match the claim.

🧩 4. Claim:

Frontier models exhibit solver diversity and exploratory capability, important for scientific discovery.

Evidence:

Section 5.7: model-wise analysis of generated numerical schemes.

Most models choose finite-difference + Euler;

DeepSeek-R1 → spectral + IMEX;

o3 → broadest scheme diversity.

Visualizations (Figures 5–6) substantiate this behavioral difference.

Assessment: ✅ Supported, though qualitative.

Diversity is clearly demonstrated; “scientific discovery” phrasing is aspirational but grounded in observed exploration behavior.

🧩 5. Claim:

LLM-generated solvers are interpretable and allow insight into errors and reasoning.

Evidence:

Reaction–Diffusion case study (§ 5.7):

The paper inspects code to show that LLMs incorrectly discretize the reaction term (missed analytical sub-solution).

This post-hoc inspection is only possible because outputs are human-readable code.

Assessment: ✅ Credible and clearly demonstrated.

Concrete example substantiates interpretability advantage over black-box neural solvers.

**Requested Changes:**

1. Clarify Reference Solvers and Ground Truth (Critical)

Although the paper lists which solvers were used for each PDE (e.g., analytical, spectral, Strang splitting), it doesn’t specify:

The numerical order, grid size, or time-stepping details of each reference.

How “ground truth” solutions were generated for error computation (analytical vs. high-resolution numerical vs. dataset-provided).

➡ Proposed adjustment: Add a concise table summarizing for each PDE family:

The reference solver type, discretization order, and grid/time step used;

How the ground truth reference is defined and validated.

This is critical for fair comparison and reproducibility.

2. Quantify Inference and Compute Costs (Critical)

The paper reports code execution runtime (solver speed), but not the total inference cost:

No data on token usage, number of LLM calls per debugging/refinement loop, or wall-clock inference time.

Test-time scaling (best-of-n) increases cost, but the trade-off is not quantified.

➡ Proposed adjustment: Include a short table summarizing inference cost per PDE family:

Average tokens or latency per solver (single-shot vs. 4-round debug vs. refined vs. best-of-n=16).

Compare total cost vs. classical or neural baselines.

This is essential for evaluating real-world feasibility.

3. Address Oracle Selection in Best-of-n Scaling (Critical)

The best-of-n scaling experiments select the lowest nRMSE among candidates, assuming access to ground truth at inference time — effectively an oracle.

➡ Proposed adjustment:
Add a variant where solver selection uses unsupervised proxies (e.g., residual norm, stability, runtime errors) instead of true nRMSE, or at least acknowledge this limitation explicitly.

This change is important for methodological transparency and external validity.

4. Include Details on Execution Environment and Safety (Important but Non-Critical)

The framework executes LLM-generated Python code, yet the paper omits details on:

Whether the runs were sandboxed,

Timeout/memory protections,

How unsafe imports or file operations were handled.

➡ Proposed adjustment:
Add a short paragraph (even in supplement) explaining code execution safety and reproducibility—e.g., “All generated solvers were executed in Docker containers with restricted I/O and 2 GB memory limits.”

This isn’t a blocker for acceptance but strengthens trust and replicability.

---

### Review · Reviewer_vcD2 · 2025-12-07

**Summary Of Contributions:**

This paper frames PDE solving as a code-generation problem and introduces CodePDE, an inference-time framework that combines (1) chain-of-thought–guided solver generation, (2) iterative automated debugging, (3) feedback-driven solver refinement, and (4) test-time scaling (best-of-n sampling) to produce executable numerical solvers from large language models. The authors evaluate 16 LLMs on five PDE families (Advection, Burgers, Reaction–Diffusion, Compressible Navier–Stokes, Darcy Flow) using metrics including normalized RMSE (nRMSE), debug success rate, empirical convergence behavior, and runtime. Key empirical findings: LLM-generated solvers with debugging + refinement outperform handcrafted reference solvers on 4 of 5 tasks after refinement; automated debugging raises average bug-free rates from 41% to 84%; test-time scaling yields steady gains up to moderate budgets (notably between n=4 and n=16); Reaction–Diffusion remains a consistent failure mode, and CNS shows high convergence-test failure rates.

**Key strengths**

- Clear framing of PDE solving as code generation and a practical, modular pipeline that unites generation, execution feedback, and refinement.

- Thorough empirical study across multiple LLMs and representative PDE families with several useful metrics (nRMSE, convergence tests, debug-success).

- Ablation evidence that debugging, refinement, and test-time scaling are individually important

**Key Weaknesses**

- The Reaction–Diffusion failure mode is identified, but the diagnosis and attempted mitigations are relatively narrow.

- Limited discussion of failure modes and when LLM-generated solvers should not be trusted

- Comparison with neural solvers and foundation models is somewhat unfair, as these methods are designed for different use cases (e.g., learning from data, zero-shot generalization)

- No analysis of the exact costs(LLM+GPU) involved while comparing different methods

**Audience:**

Yes

**Audience Explanation:**

This paper addresses an important intersection of machine learning and scientific computing that is highly relevant to the TMLR audience

**Claims And Evidence:**

Yes

**Claims Explanation:**

The paper provides substantial empirical evidence to support its main claims

**Requested Changes:**

The additions below would further strengthen the work:

- Discuss when LLM-generated solvers should not be trusted and their failure cases

- Clarify the fairness and scope of comparisons with neural PDE solvers and foundation models (Add an explicit section in the Appendix/Discussions to discuss the fundamental differences between PDE solvers and LLMs)

- Provide a transparent accounting of compute and monetary costs for each method (Exact LLM inference costs, GPU hours etc)

---

### Comment · Action_Editor_bw6F · 2025-12-07
**Respond to reviews**

Dear authors:

Thank you for your submission. Now that all three reviews are public and the discussion phase has started, please note that you have up to two weeks from the start of the discussion phase to post your author response. We encourage you to address all reviewer comments and questions as clearly and concretely as possible.

---

> ### Author Response · Authors · 2025-12-20
>
> We thank the action editor for handling our paper. We have responded to the reviewers and updated our paper accordingly. The updated parts in the revision have been highlighted in $\text{\color{red}red}$. We look forward to discussing with reviewers!

---

### Decision · Action_Editor_bw6F · 2026-02-11

**Recommendation:** Accept as is

**Audience:**

Yes

**Audience Explanation:**

AI4Science community will be interested in reading this paper.

**Claims And Evidence:**

Yes

**Claims Explanation:**

This paper makes a good contribution by formulating PDE solving as executable code generation and presenting an inference-time pipeline (generation, automated debugging, feedback-driven refinement, and test-time scaling) that substantially improves solver correctness and accuracy. The empirical evaluation spans 16 LLMs, five diverse PDE families, multiple metrics, and convincingly demonstrates that debugging and best-of-n sampling deliver consistent gains with several tasks exceeding reference baselines.